# Terrain Self-Similarity-Based Transformer for Generating Super Resolution DEMs

Xin Zheng , Zelun Bao and Qian Yin *

School of Artificial Intelligence, Beijing Normal University, Beijing 100875, China
* Correspondence: yinqian@bnu.edu.cn

**Abstract:** High-resolution digital elevation models (DEMs) are important for relevant geoscience research and practical applications. Compared with traditional hardware-based methods, super-resolution (SR) reconstruction techniques are currently low-cost and feasible methods used for obtaining high-resolution DEMs. Single-image super-resolution (SISR) techniques have become popular in DEM SR in recent years. However, DEM super-resolution has not yet utilized reference-based image super-resolution (RefSR) techniques. In this paper, we propose a terrain self-similarity-based transformer (SSTrans) to generate super-resolution DEMs. It is a reference-based image super-resolution method that automatically acquires reference images using terrain self-similarity. To verify the proposed model, we conducted experiments on four distinct types of terrain and compared them to the results from the bicubic, SRGAN, and SRCNN approaches. The experimental results show that the SSTrans method performs well in all four terrains and has outstanding advantages in complex and uneven surface terrains.

**Keywords:** DEM; super-resolution reconstruction; transformer; self-similarity





## 1. Introduction

A digital elevation model (DEM) is a digital terrain simulation enabled by topographic elevation data. DEMs can provide precise geographic information and are increasingly being used in fields such as hydrology, ecology, meteorology, and topographic mapping [1–5]. High-resolution DEMs are more detailed and can provide more accurate representations of terrain surfaces; DEM quality is essential for relevant geoscientific research and real-world applications. For example, the findings of flood model simulations demonstrate that DEM accuracy can significantly affect flood danger estimation [6], and DEM accuracy is practically linearly proportional to terrain slope, i.e., the steeper the slope, the higher the error [7]. The main sources for generating DEMs are GPS and remote sensing [8]. Among the remote sensing methods, LiDAR techniques have contributed significantly to the acquisition of high-resolution DEMs [9]. However, creating DEMs with LiDAR methods is expensive, and obtaining high-quality DEMs over a wide area is a challenge. Therefore, a feasible strategy to obtain high-resolution DEMs at a low cost is to use super-resolution reconstruction techniques to reconstruct low-resolution DEMs into high-resolution DEMs [10].

Traditional interpolation methods, such as inverse distance weighting (IDW), bilinear interpolation, nearest-neighbor interpolation (NNI), and bicubic interpolation [11–15] were widely used in the early work, but these methods are susceptible to terrain relief, resulting in less stable accuracy [16,17]. The approach of fusing multiple data sources to construct a high-resolution DEM is also frequently utilized [18–20]. Although this method can use the complementary qualities of multi-source data to extract significant information from them, it can not significantly increase the accuracy of the rebuilt data in the case of limited data sources.

Single-image super-resolution (SISR) and reference-based image super-resolution (RefSR) are the two basic approaches used in deep learning-based super-resolution (SR)

research. The SISR technique has been widely used for DEM SR reconstruction. Enormous quantities of DEM sample data are used by the deep learning-based DEM SR reconstruction method to learn how to rebuild a low-resolution DEM and produce a high-resolution DEM that accurately represents the terrain [21–23]. The first method to reconstructing high-resolution images using convolutional neural networks is called super-resolution convolutional neural networks (SRCNNs) [24]. Chen et al. [25] used SRCNN to DEM scenes (D-SRCNN) and achieved superior reconstruction results compared to traditional interpolation approaches. By incorporating gradient information into a depth super-resolution network (EDSR) via transfer learning, Xu et al. [22] produced high-quality DEMs while resolving the issues of enormous dynamic height ranges and inadequately trained samples. Demiray et al. [26] developed a D-SRGAN model based on generative adversarial networks (GANs) for enhancing DEM resolution, inspired by the SISR technique. Zhu et al. [27] presented a conditional encoder–decoder generative adversarial neural network (CEDGAN) for DEM that captures the complicated features of the input's spatial data distribution, which was inspired by conditional generative adversarial networks (CGANs) [28].

Compared to SR reconstruction without a reference image, SR reconstruction using a reference image can provide more detailed information and hence achieve superior reconstruction results. Recently, advances have been made in RefSR, which transfers high-resolution information from a specific reference picture to generate satisfying results [29–31]. Yue et al. [29] proposed a more general scheme using reference images, in which similar images are retrieved from the web, and globally registered and locally matched. To apply semantic matching, Zheng et al. [30] substituted convolutional neural network features for the straightforward gradient features and employed a SISR approach for feature synthesis. Yang et al. [31] were among the first to introduce the transformer architecture in SR tasks.

The current super-resolution image reconstruction method without a reference image is widely used on DEM data. However, the super-resolution image reconstruction method with a reference image is not used in DEM high-resolution reconstruction because the manual method of providing a reference image is difficult to implement. Inspired by Zheng's SWM method [10], which extracts self-similarity from an input image to generate a high-resolution image, we propose a method for automatically obtaining reference data for low-resolution DEM data by utilizing terrain self-similarity in this paper. In mathematics, a self-similar object is exactly or nearly similar to a part of itself. Self-similarity has also been verified in geographic phenomena [32–34]. The self-similarity of the terrain can be used to construct reference images that have greater information and generate superior results than single-image super-resolution methods. We are one of the first to introduce the RefSR into DEM SR. In addition, we apply a transformer model for image super-resolution inspired by Yang's TTSR approach [31], where low-resolution (LR) and reference(s) (Ref) correspond to the query and key in the transformer [35], respectively.

The structure of this paper is as follows. The application and gathering of DEM data as well as associated research on DEM super-resolution are discussed in Section 1. The dataset and model construction processes are thoroughly explained in Section 2. Data sources, experiments, and analysis of experimental results are all described in Section 3. Finally, we discuss the conclusions of the paper and possible future research directions in Section 4.

## 2. Methodology

The following are the main steps of the experiment as shown in Figure 1.

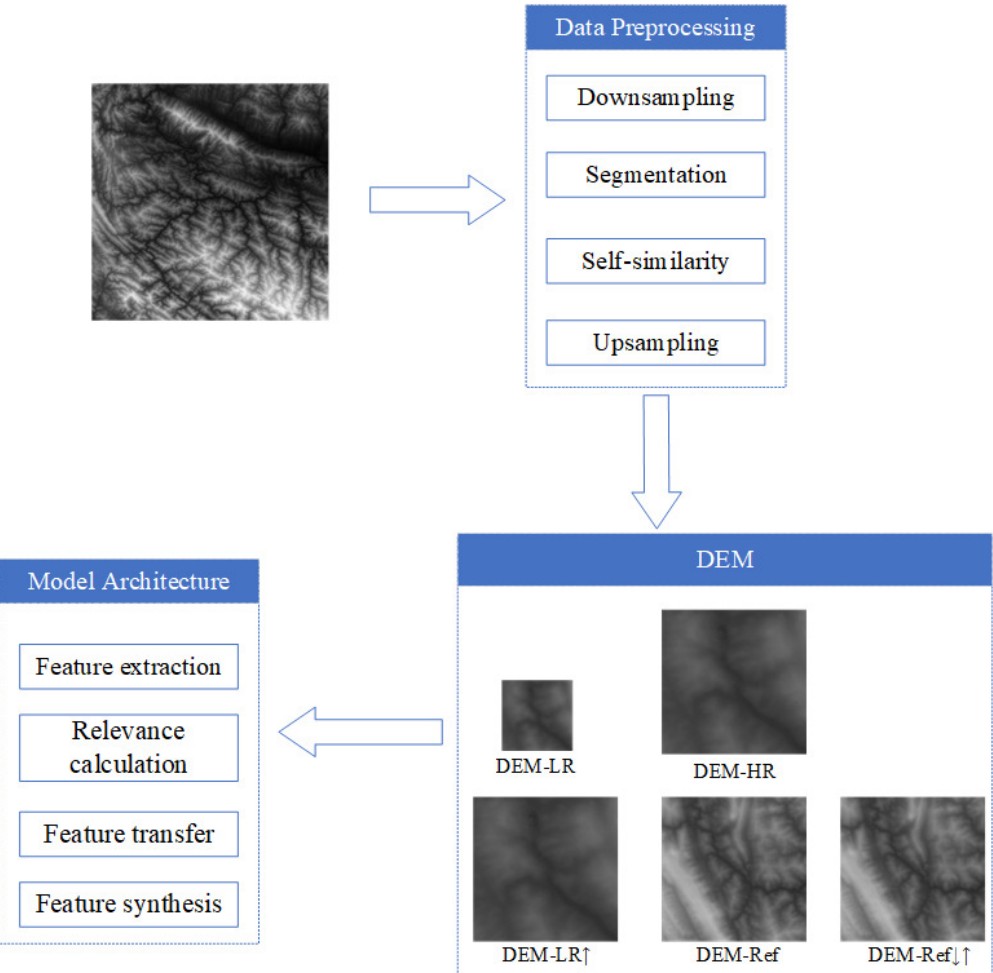

**Figure 1.** The self-similarity transformer workflow. DEM-HR refers to the high-resolution DEM data used for comparison with SR data. DEM-LR refers to the low-resolution DEM data obtained after downsampling DEM-HR as input. DEM-Ref refers to the reference data obtained using self-similarity. DEM-LR↑ refers to the data obtained by upsampling DEM-LR, while DEM-Ref↓↑ refers to the data obtained by downsampling and upsampling the reference data. DEM-LR↑ and DEM-Ref↓↑ will be used to calculate the correlation between the low-resolution image and the reference image.

### 2.1. Self-Attention in Transformers

The original transformer model was used in natural language processing. Transformer networks have received great interest in computer vision due to their excellent performance in natural language processing. As a result, the transformer model has been extensively studied in the field of image super-resolution [36,37]. The foundation of the transformer design is a self-attention mechanism that picks up on the connections between the elements. In the original transformer model, $X$ represents a sequence of $n$ entities $(x_1, x_2, \dots, x_n)$. The self-attention formula can be expressed as:

$$\text{Attention}(V, K, Q) = \text{softmax}(\frac{QK^T}{\sqrt{d_k}})V \tag{1}$$

where $V = XW^V, K = XW^K, Q = XW^Q, W^V, W^K, W^Q$ represents three learnable weight matrices to transform $V$(value), $K$(key), $Q$(query); $d_k$ represents the dimension of the query and key.

### 2.2. TTSR

TTSR [31] is the first method to use a transformer structure in image super-resolution and has achieved significant improvement. TTSR employs an image super-resolution

method based on reference image(s) (RefSR), where the transformer's representations of the LR and Ref images are used as query and key, respectively. This architecture enables learning to combine features from the LR and Ref images to identify deep-matching features.

### 2.3. Data Pre-Processing

Figure 2 shows the data preprocessing process. First, the original high-resolution DEM is cropped to obtain an N × N sized DEM-HR image. This image is used as the ground truth for comparison with the DEM-SR results. Then, we apply bicubic downsampling with a factor of a× on DEM-HR to obtain DEM-LR, which is used as the super-resolution input image. Unlike the traditional reference-based SR method that uses high-resolution images as reference images, there is no available reference terrain dataset, and constructing the dataset is a huge workload. Considering the self-similarity of the terrain, we use large-scale DEM images centered on DEM-LR as reference images and apply them to super-resolution reconstruction. With DEM-HR as the center, we crop the original high-resolution DEM to obtain a DEM image of size a*N × a*N and then apply bicubic downsampling with the factor a× on the a*N × a*N DEM image to obtain DEM-Ref, which is used as the reference image of DEM-LR. Then we apply bicubic upsampling with the factor a× on DEM-LR to obtain a DEM-LR↑ image and we sequentially downsample and upsample the DEM-Ref with the same factors a× to obtain DEM-Ref↓↑. The correlation between the LR and the reference could be calculated via the use of the DEM-LR↑ and DEM-Ref↓↑. Finally, our model generates a synthetic feature map from the inputs of DEM-LR, DEM-Ref, DEM-LR↑, and DEM-Ref↓↑, and uses it to produce HR predictions.

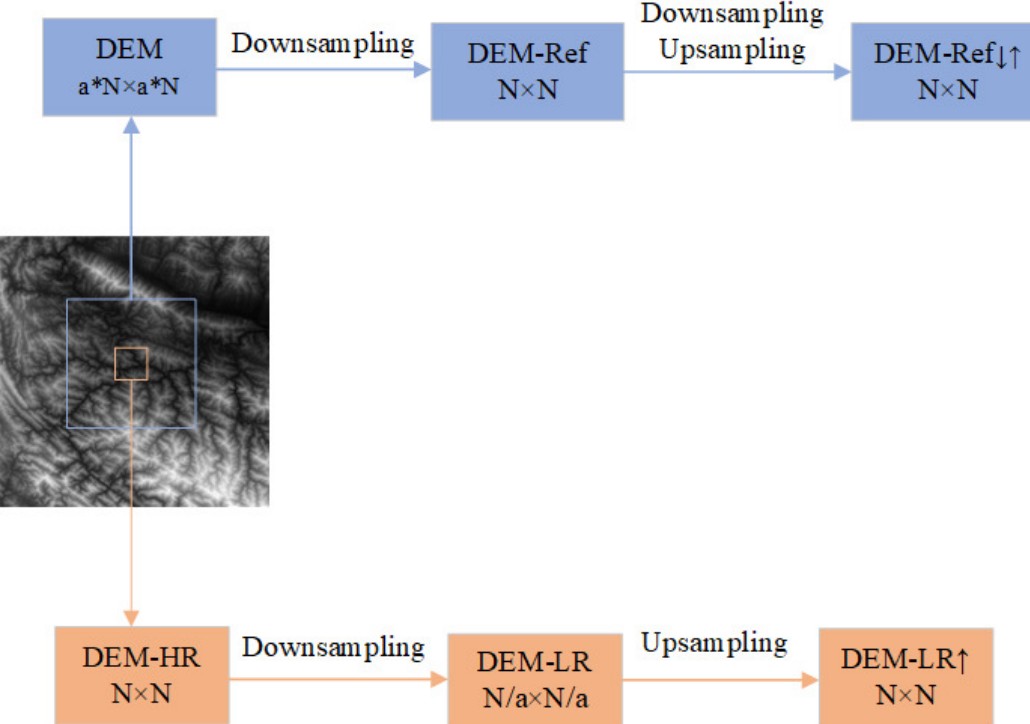

**Figure 2.** Data preprocessing flow. DEM-HR refers to the high-resolution DEM data used for comparison with SR data. DEM-LR refers to the low-resolution DEM data obtained after downsampling DEM-HR as input. DEM-Ref refers to the reference data obtained using self-similarity. DEM-LR↑ refers to the data obtained by upsampling DEM-LR, while DEM-Ref↓↑ refers to the data obtained by downsampling and upsampling the reference data. We will use DEM-LR↑ and DEM-Ref↓↑ to calculate the correlation between the low-resolution image and the reference image.

*2.4. Model Architecture*

2.4.1. Self-Similarity Transformer

As shown in Figure 3, there are four parts in our transformer: residual feature extraction module, relevance calculation module, feature transfer module, and feature synthesis module.

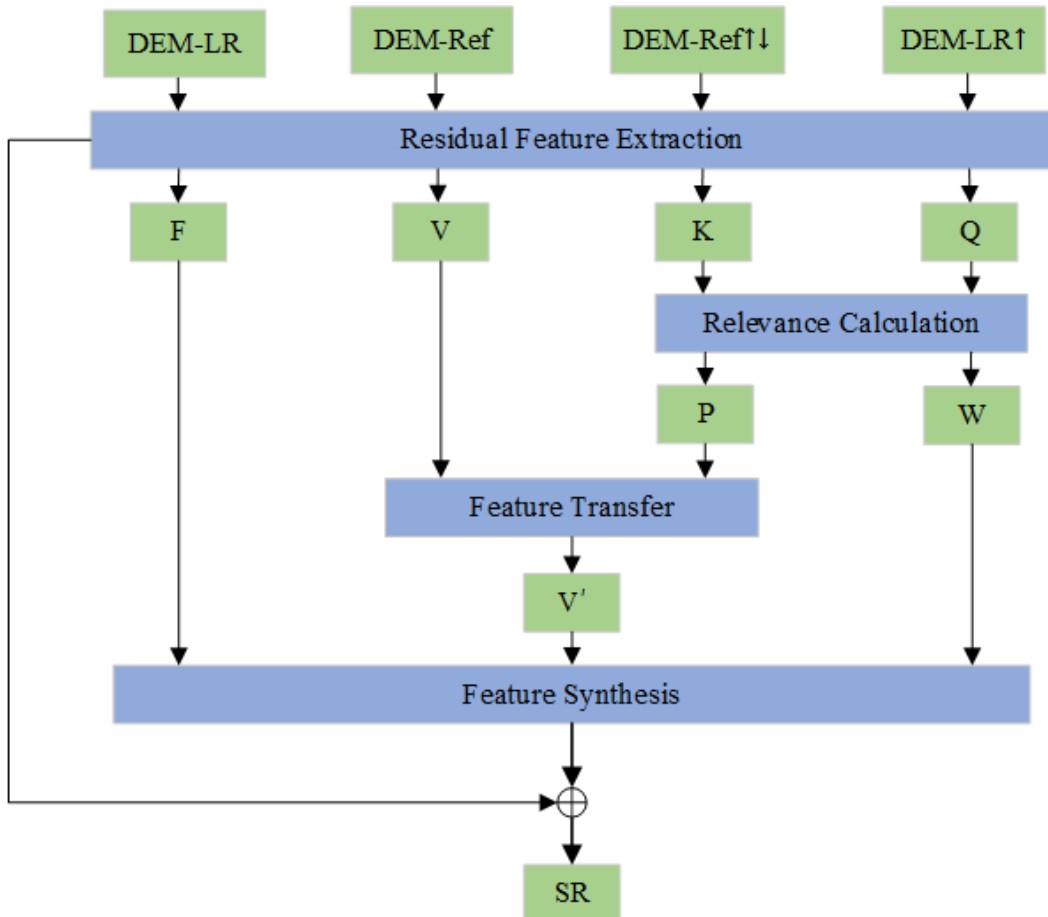

**Figure 3.** SSTrans structure. F, V, K, Q are the features of DEM-LR, DEM-Ref, DEM-Ref↓↑, and DEM-Ref extracted by the residual network. P, W are the position matrix and weight matrix obtained by the correlation calculation, respectively. $V'$ is the high-resolution feature representation of DEM-LR.

Accurate and appropriate feature extraction for reference images is helpful for generating better high-resolution images. We use a residual network-based feature extraction method. Through the combination of LR and Ref image feature learning, this approach may produce more precise similar features. The feature extraction procedure can be described as follows:

$$F = \text{RFE}(DEM - LR) \tag{2}$$

$$V = \text{RFE}(DEM - Ref) \tag{3}$$

$$K = \text{RFE}(DEM - Ref \downarrow\uparrow) \tag{4}$$

$$Q = \text{RFE}(DEM - LR \uparrow) \tag{5}$$

where $\text{RFE}(\cdot)$ denotes the residual feature extraction module. Features $V$ (value), $K$ (key), and $Q$ (query) correspond to the three basic components of the attention mechanism within the transformer, and $F$ is a DEM-LR feature.

Calculating the similarity between $Q$ and $K$ yields the correlation between DEM-LR and DEM-Ref. The relevance calculation operation was used to record the position information most relevant to the DEM-LR image in the DEM-Ref image:

$$P, W = RC(Q, K) \tag{6}$$

where $RC(\cdot)$ denotes the relevance calculation operation, which uses element-wise multiplication. $W$ is the relevance weight matrix; $P$ is the relevance position matrix.

Through the position matrix $P$, the features of $V$ are transferred to obtain the representation of HR features corresponding to DEM-LR images:

$$V' = FT(V, P) \tag{7}$$

where $FT(\cdot)$ denotes the feature transfer operation, which uses hard attention [31]. $V'$ represents the HR feature representation for the DEM-LR image.

We synthesize features $F$, $V'$, and $W$ to obtain the final output result. This method can be defined as follows:

$$FS(F, V', W) = Conv(F||V') \odot W \tag{8}$$

$$I^{SR} = F + FS(F, V', W) \tag{9}$$

where $I^{SR}$ indicates the synthesized output features, $FS(\cdot)$ denotes the feature synthesis operation, the operator $\odot$ denotes the Hadamard product between feature maps, $||$ denotes channel-wise concatenation, and $Conv$ denotes a convolutional layer.

### 2.4.2. Loss Function

Our loss function adopts adversarial loss $\mathcal{L}_{adv}$ and reconstruction loss $\mathcal{L}_{rec}$. Adversarial loss could improve the visual quality of synthetic pictures greatly [38,39]. We use WGAN-GP [40] to obtain more stable results. The adversarial loss is expressed as:

$$\mathcal{L}_{adv} = -_{\tilde{x} \sim_g}[D(\tilde{x})] \tag{10}$$

$$\min_{G} \max_{D \in \mathcal{D}} {}_{x \sim_r}[D(x)] - {}_{\tilde{x} \sim_g}[D(\tilde{x})] \tag{11}$$

In this paper, we use the L1 loss as our reconstruction loss instead of the mean square error measure (MSE).

$$\mathcal{L}_{rec} = \left\| I^{HR} - I^{SR} \right\|_1 \tag{12}$$

### 2.4.3. Implementation Details

The weights for $\mathcal{L}_{rec}$ and $\mathcal{L}_{adv}$ are 1 and $1 \times 10^{-4}$, respectively. The Adam optimizer is used with the learning rate of $1 \times 10^{-4}$ The network is pre-trained for 2 epochs, where only $\mathcal{L}_{rec}$ is applied. Afterward, all losses are used to train for another 60 epochs.

### 2.5. Evaluation Metrics

The root mean square error (RMSE) and mean absolute error (MAE) are frequently employed as markers to assess the accuracy of the reconstruction. The quality of the reconstruction improves as the MAE and RMSE absolute values decrease.

$$MAE = \frac{1}{N} \sum_{i}^{N} |y_i - y'_i| \tag{13}$$

$$RMSE = \sqrt{\frac{1}{N} \sum_{i}^{N} (y_i - y')^2} \tag{14}$$

where $N$ denotes the number of pixels in the DEM sample, the value of each pixel in the original data are represented by $y_i$, and the value of each pixel in the reconstruction result is represented by $y'_i$.

In addition, in this study, we use structural similarity (SSIM) [41] and peak signal-to-noise ratio (PSNR) to assess the similarity of the terrain to one another.

$$PSNR = 10 \times \log_{10}(\frac{(2^n - 1)^2}{MSE}) \tag{15}$$

where $MSE$ is the mean square error.

The mean errors of the terrain parameters are represented by $E_{tp}$.

$$E_{tp} = \frac{1}{N} \sum_i^N |t_i - t'_i| \tag{16}$$

where $t_i$ denotes the values of the terrain parameters generated with the original high-resolution DEM and $t'_i$ denotes the values of the terrain parameters generated with the reconstructed high-resolution DEM.

### 3. Experiments and Results

The experiment uses the Ubuntu 16.04 operating system, an Intel Xeon Gold 6230 CPU, a Tesla V100 SXM2 GPU with 32 GB, and networks built using the PyTorch framework. Bicubic, SRCNN, and SRGAN are the three super-resolution reconstruction algorithms that are used as comparisons in comparative studies to assess the effectiveness of the new approaches suggested. SRCNN and SRGAN are part of the single-image super-resolution (SISR) approach, while bicubic belongs to the traditional interpolation algorithm. Additionally, the MSE, RMSE, PSNR, and SSIM assessment metrics are used to evaluate the four super-resolution reconstruction algorithms.

### 3.1. Data Descriptions

The experimental data used in this work were provided by the ASTER GDEM V3 dataset with a data resolution of 30 m. Figure 4 shows four typical subregions of mainland China, i.e., the Inner Mongolian Plateau, the Tarim Basin, the Qinling Mountains, and the North China Plain, which were selected as the ground truth to evaluate our model's performance. These areas comprise a variety of terrains with a wide range of hypsography and altitude. According to these four areas, we built a DEM dataset based on the self-similarity of the terrain. A total of 40,000 DEM pairs form the DEM dataset, of which 30,000 pairs form the training set and 10,000 pairs form the validation set. Each subarea contains 10,000 DEM pairs, of which, 7500 are used for training and 2500 for validation. Each pair contains an input image and a reference image, and the parameters N and a in Section 2.3 are set to 32 and 4, respectively. The input image is a 32 × 32 DEM cropped from the original high-resolution DEM data, the reference image is a 32 × 32 DEM obtained by using terrain self-similarity, corresponding to the input image.

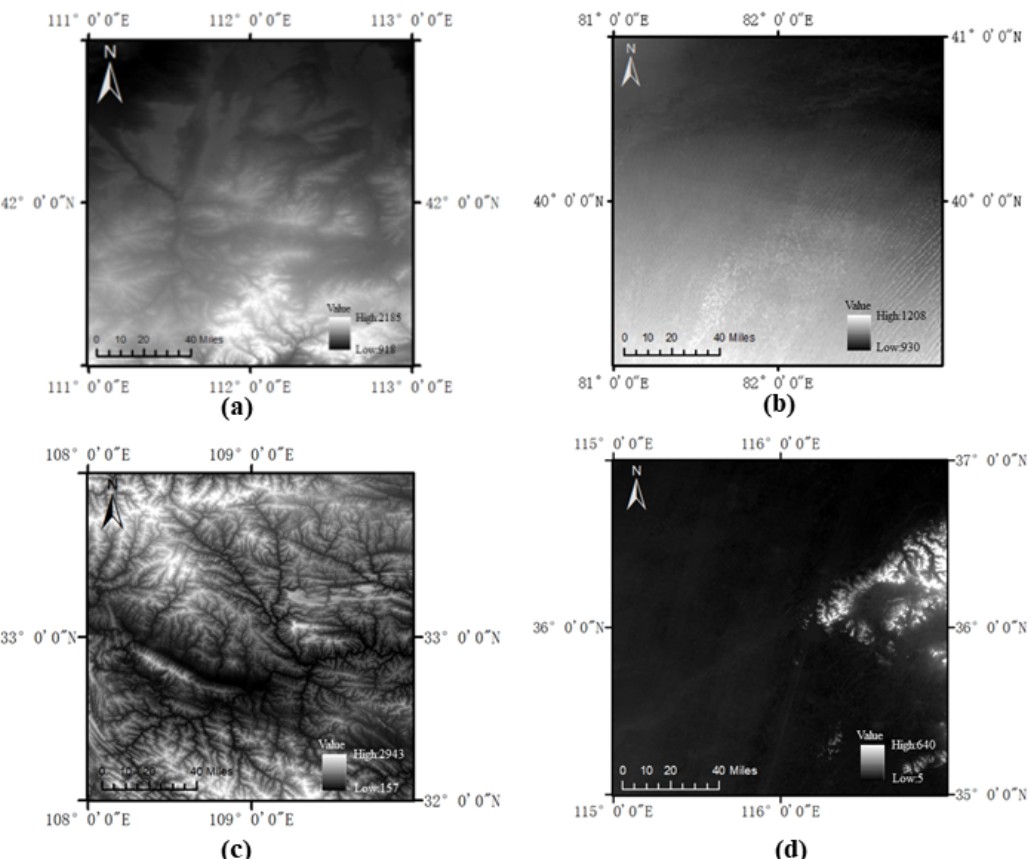

**Figure 4.** (**a**) Inner Mongolian Plateau, (**b**) Tarim Basin, (**c**) Qinling Mountains, (**d**) North China Plain.

### 3.2. Results of the SR in Four Test Areas

To verify the reconstruction effect in DEM of the proposed model, 900 × 900 DEMs of the Inner Mongolian Plateau, Qinling Mountains, Tarim Basin, and North China Plain were selected, and the maximum elevation differences of the four regions are shown in Table 1. Figure 5 shows the results of super-resolution reconstruction. Here are the conclusions of the experiment:

Area 1 is located in the Inner Mongolia Plateau, with high terrain and a relatively smooth surface. As shown in Table 1, area 1 has a maximum elevation of 2206 m and a minimum elevation of 1260 m, with a maximum elevation difference of 946 m. Figure 5(b1) shows how closely the experimental results to the original DEM are reconstructed. As shown in Table 2, due to the large height difference, the MAE and RMSE values are relatively large, with a MAE value of 4.44 and an RMSE value of 5.65. The PSNR value is 34.09 and the SSIM value is 98.93%. The experiments demonstrate that the reconstruction results are highly similar to the original DEM data and have small errors.

**Table 1.** Maximum elevation differences in four areas.

| Area | Maximum Elevation (m) | Minimum Elevation (m) | Maximum Elevation Difference (m) |
|---|---|---|---|
| Area 1 | 2206 | 1260 | 946 |
| Area 2 | 2528 | 190 | 2338 |
| Area 3 | 1109 | 906 | 203 |
| Area 4 | 129 | 5 | 124 |

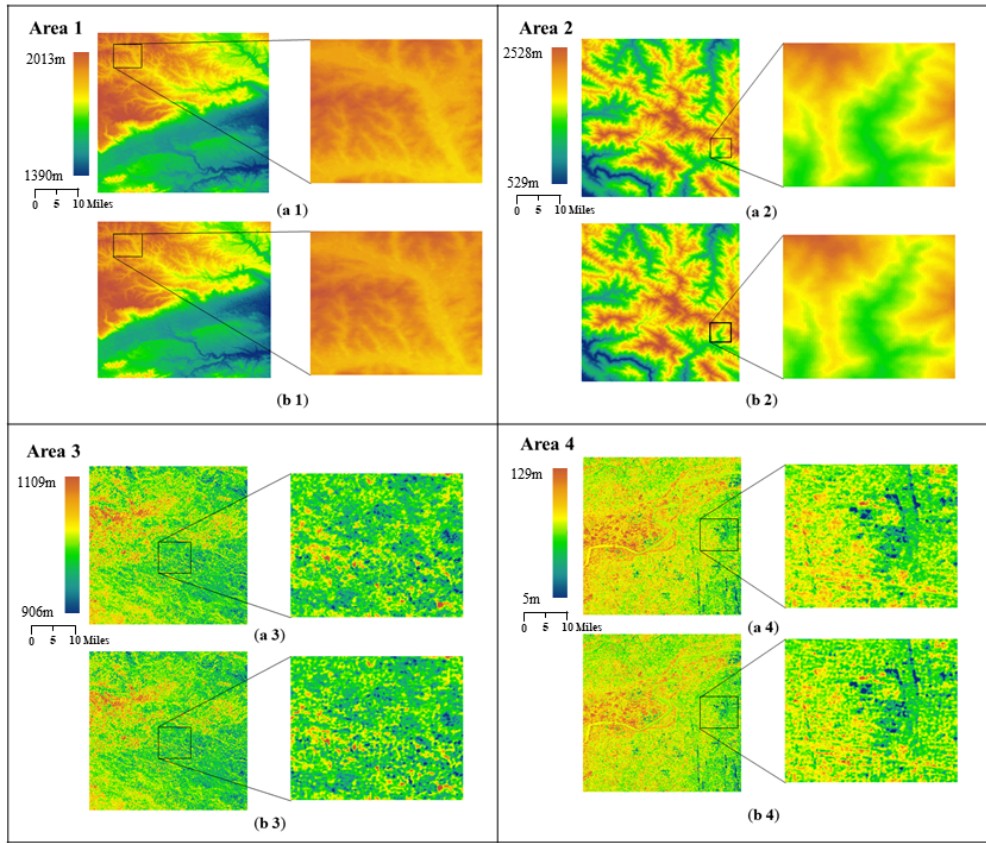

**Figure 5.** DEM reconstruction visualization results: (**a1–a4**) is the original DEM, (**b1–b4**) is the reconstruction DEM.

Area 2 is located in the Qinling Mountains; the topography of the area is relatively simple but the terrain is highly variable. As shown in Table 1, area 2 has a maximum elevation of 2528 m and a minimum elevation of 190 m, with a maximum elevation difference of 2338 m. Figure 5(b2) shows how closely the experimental results to the original DEM are reconstructed. As shown in Table 2, due to the significant topographic variation, MAE and RMSE values are 12.96 and 16.52, respectively. However, the reconstructed similarity is very high, with SSIM values as high as 99.04%.

**Table 2.** MAE, RMSE, PSNR, and SSIM values of the reconstructed effects in four areas.

| Area | MAE (m) | RMSE (m) | PSNR (dB) | SSIM |
|---|---|---|---|---|
| Area 1 | 4.44 | 5.65 | 34.09 | 98.93% |
| Area 2 | 12.96 | 16.52 | 23.77 | 99.04% |
| Area 3 | 1.55 | 2.03 | 41.99 | 96.13% |
| Area 4 | 1.63 | 2.14 | 41.51 | 94.11% |

Area 3 is located in the Tarim Basin, with high terrain and a non-smooth surface. As shown in Table 1, area 3 has a maximum elevation of 1109 m and a minimum elevation of 906 m, with a maximum elevation difference of 203 m. Figure 5(b3) shows how closely the experimental results to the original DEM are reconstructed. In Table 2, the MAE value is 1.55 and the RMSE value is 2.03. The PSNR value is 41.99, which is higher than the values in regions 1 and 2, and the SSMI value is 96.13%, which is lower than the values in regions 1 and 2.

Area 4 is located in the North China Plain, with relatively complex topographic texture features and a non-smooth surface. As shown in Table 1, area 4 has a maximum elevation of 129 m and a minimum elevation of 5 m, with a maximum elevation difference of 124 m. Figure 5(b4) shows how closely the experimental results to the original DEM are

reconstructed. The MAE value is 1.63, the RMSE value is 2.14, the PSNR value is 41.54, and the SSIM value is 94.11%, as shown in Table 2.

Based on the reconstruction results of the four areas, we can conclude that our model can achieve effective reconstruction results, with a maximum regional structural similarity (SSIM) of terrain smoothing index of 99%; the greater the height difference of the terrain, the greater the MAE and RMSE values.

### 3.3. Comparison Analysis with Other SR Methods

A comparative analysis using bicubic interpolation, SRGAN, and SRCNN was performed to confirm the superiority of the models. The comparison of different methods is shown in Table 3 and Figure 6.

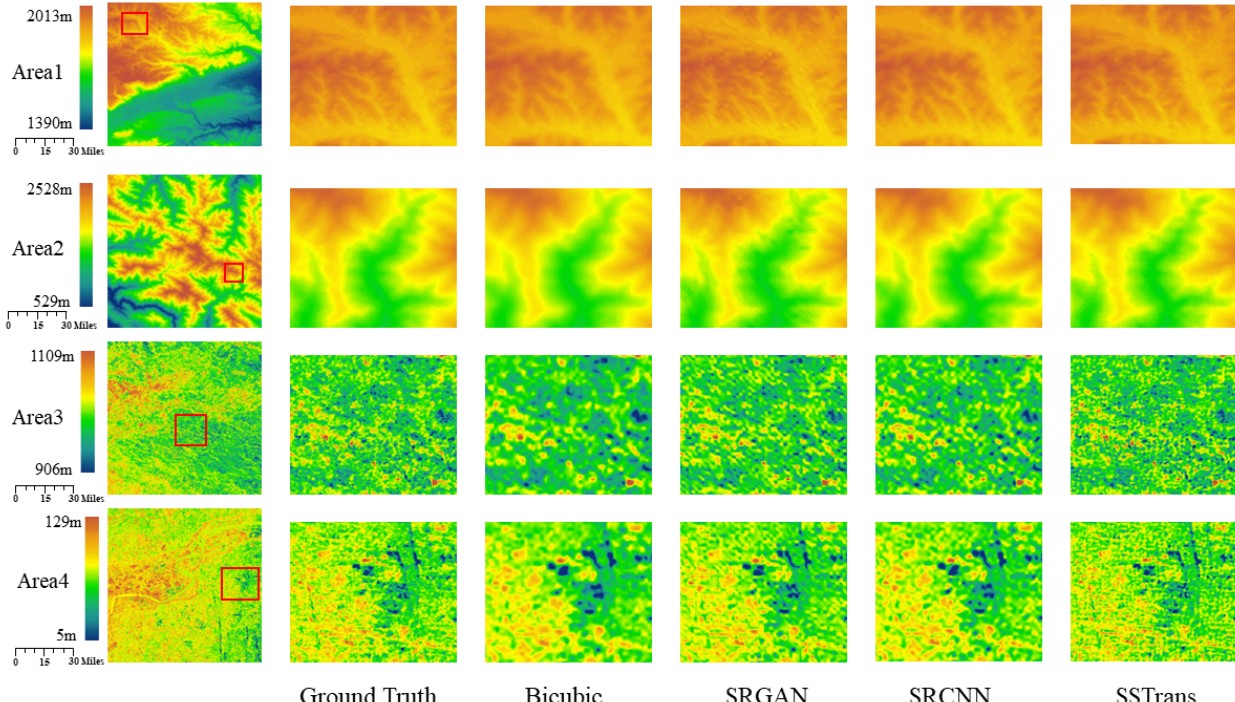

**Figure 6.** Comparison of DEM reconstruction visualization results of four methods in four areas.

Both areas 1 and 2 have relatively large elevation differences; the MAE and RMSE values obtained by all four methods are relatively large and the PSNR values are small. Therefore, the size of the elevation difference is an important factor affecting the reconstruction results. The topographic surfaces of areas 1 and 2 are smoother, and the SSIM values obtained by all four methods are high above 95%; Figure 6 also shows that the reconstructed results of area 1 and area 2 have high similarity. Areas 3 and 4 have more complex terrain surfaces where the advantages of SSTrans are more evident. For example, in area 4, the MAE value of SSTrans is 32.08% lower, the RMSE value is 32.49% lower, the PSNR value is 8.95% higher, and the SSMI value is 15.76% higher compared to the SRCNN. As shown in Figure 6, SSTrans still maintains high-quality reconstruction in areas 3 and 4, with SSIM values above 90%. Compared to the other SR methods, the SSTrans method achieved the best results in all four areas.

Figure 7 shows the histogram statistics of the frequency of elevation points for regions 3 and 4. In general, the SSTrans method is very close to the original DEM. Figure 8 displays the results of the four-regional hillshade for a better visual assessment of the terrain relief. As a traditional interpolation method, bicubic is unreliable and ineffective in recovering details. SRGAN and SRCNN can recover more detailed information, but in areas where the terrain surface is very complex, such as regions 3 and 4, they cannot accurately reconstruct the terrain, and the difference with the original DEM is relatively

large. SSTrans complements the shortcomings of SRCNN and SRGAN by taking advantage of the self-similarity of the terrain to obtain more information from the reference DEM, thereby reconstructing the details in the DEM more accurately.

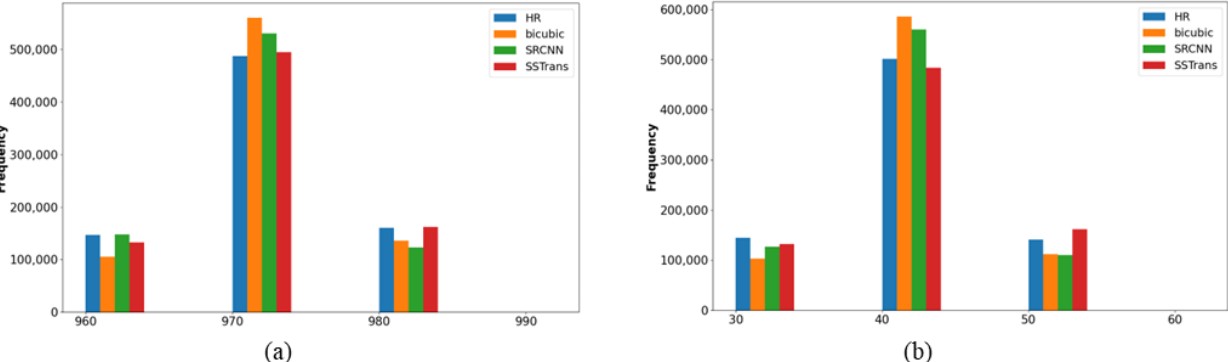

(a) (b)

**Figure 7.** Histogram statistics of the DEM reconstruction, (**a**) represents area 3 and (**b**) area 4.

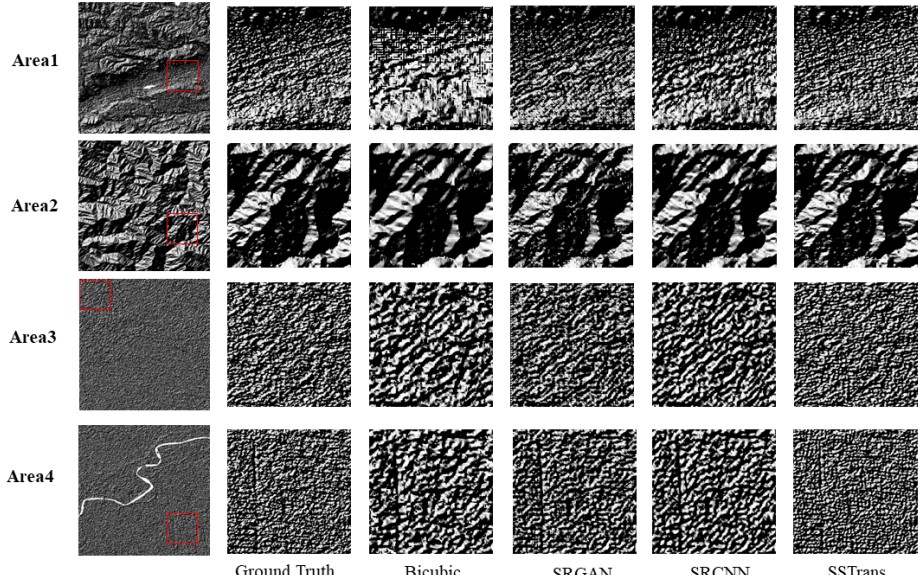

**Figure 8.** A comparison of the results of DEM reconstruction using hillshade visualization for the four methods.

**Table 3.** Quantitative evaluation of the reconstruction effects in four areas.

| Area | Methods | MAE (m) | RMSE (m) | PSNR (dB) | SSIM |
|---|---|---|---|---|---|
| Area 1 | Bicubic | 6.12 | 7.30 | 30.47 | 97.31% |
| | SRGAN | 6.17 | 8.15 | 29.90 | 96.09% |
| | SRCNN | 5.02 | 6.26 | 31.91 | 98.38% |
| | SSTrans | **4.44** | **5.65** | **33.06** | **98.93**% |
| Area 2 | Bicubic | 15.24 | 19.28 | 21.39 | 98.21% |
| | SRGAN | 17.79 | 23.10 | 20.86 | 97.54% |
| | SRCNN | 14.86 | 18.20 | 22.02 | 98.85% |
| | SSTrans | **12.96** | **16.52** | **23.77** | **99.04**% |
| Area 3 | Bicubic | 2.46 | 3.18 | 38.08 | 86.32% |
| | SRGAN | 2.10 | 2.78 | 39.22 | 87.71% |
| | SRCNN | 2.22 | 2.87 | 38.99 | 89.37% |
| | SSTrans | **1.55** | **2.03** | **41.99** | **96.13**% |
| Area 4 | Bicubic | 2.53 | 3.32 | 37.07 | 74.76% |
| | SRGAN | 2.48 | 3.29 | 37.79 | 77.08% |
| | SRCNN | 2.40 | 3.17 | 38.10 | 78.35% |
| | SSTrans | **1.63** | **2.14** | **41.51** | **94.11**% |

### 3.4. Terrain Parameters Maintenance

Table 4 shows the accuracy of the reconstruction of the slope direction and slope in the four areas; SSTrans achieved the best results compared to the other three methods. In areas 1 and 2, the slope errors of SSTrans were 35.73 and 21.86% lower than those of SRCNN; for regions 1 and 2, the aspect errors of SSTrans were 37.81% and 22.78% lower than those of SRCNN. In areas 3 and 4, the slope errors of SSTrans were 46.92% and 34.92.86% lower than those of SRCNN; for regions 1 and 2, the aspect errors of SSTrans were 57.99% and 59.70% lower than those of SRCNN. The terrain surface is more complex in areas 3 and 4 compared to areas 1 and 2, and SSTrans is further enhanced in areas 3 and 4 with far better results than other methods, especially in the aspect terrain parameter.

**Table 4.** Quantitative evaluation of terrain parameter retention in four areas.

| Area | Terrain Parameters | Bicubic | SRGAN | SRCNN | SSTrans |
|---|---|---|---|---|---|
| Area 1 | $E_{slope}$ | 3.30 | 4.07 | 3.05 | 1.96 |
|        | $E_{aspect}$ | 68.11 | 75.39 | 63.97 | 39.78 |
| Area 2 | $E_{slope}$ | 5.28 | 7.66 | 5.17 | 4.04 |
|        | $E_{aspect}$ | 29.60 | 42.39 | 28.71 | 22.17 |
| Area 3 | $E_{slope}$ | 2.50 | 2.13 | 2.11 | 1.12 |
|        | $E_{aspect}$ | 84.41 | 86.05 | 79.25 | 33.29 |
| Area 4 | $E_{slope}$ | 2.93 | 2.42 | 2.52 | 1.64 |
|        | $E_{aspect}$ | 86.99 | 87.07 | 83.74 | 33.75 |

The visualization results of the DEM reconstruction for slope and aspect using each of the four algorithms are displayed in Figures 9 and 10. The color differences between Figures 9a and 10a and the other image series show the ability of different methods to maintain terrain features. In Figures 9c and 10c, the traditional interpolation method bicubic results in large color blocks and fails to recover detailed information. In comparison, in Figures 9d,e and 10d,e, the deep learning methods, SRGAN and SRCNN, perform better and recover more detailed information, but there is still some gap with the original DEM data and they do not perform well in some finer details. In Figures 9f and 10f, the SSTrans method has further improved the results and is already very close to the original DEM in terms of the visual aspect.

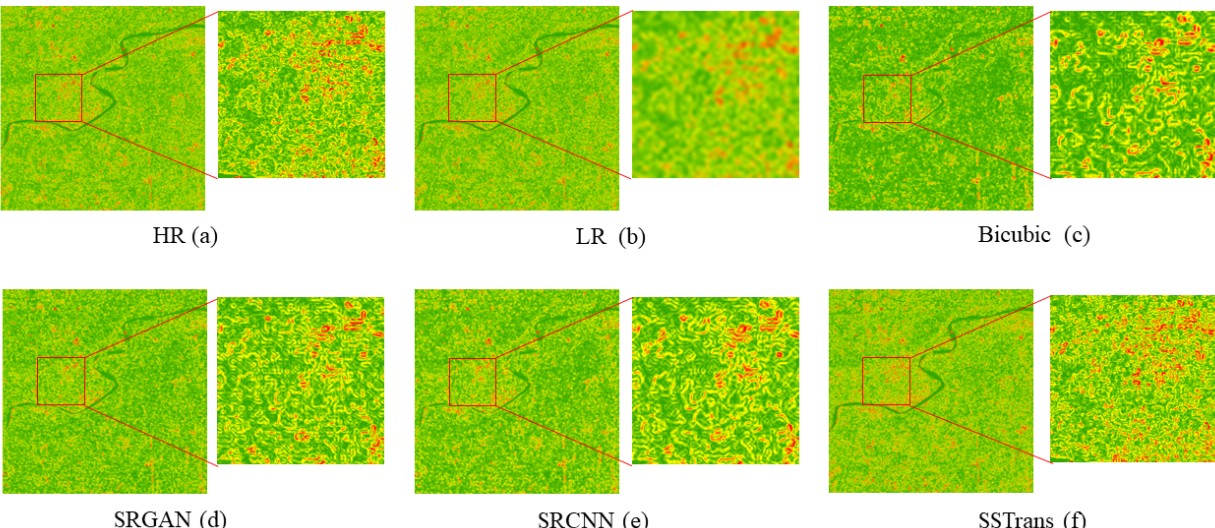

HR (a)   LR (b)   Bicubic (c)

SRGAN (d)   SRCNN (e)   SSTrans (f)

**Figure 9.** Comparison of the results of the slope visualization for DEM reconstruction using four different methods in area 4.

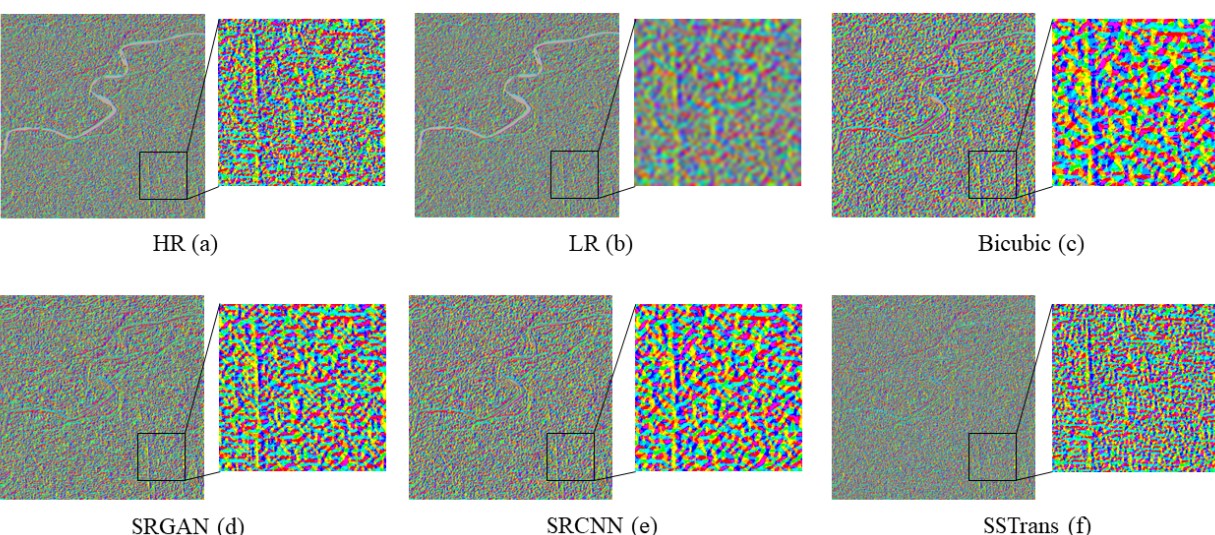

**Figure 10.** Comparison of the DEM reconstruction results for aspect visualization using four methods in area 4.

## 4. Conclusions

In this article, we propose a terrain self-similarity-based transformer for super-resolution DEM generation. The novelty of this paper is as follows.

1.  We are one of the first to introduce the transformer method to DEM super-resolution (SR);
2.  We are one of the first to introduce the reference-based image super-resolution (RefSR) into DEM super-resolution (SR);
3.  To overcome the problem that the manual method of providing reference images is difficult to implement, we propose a method to automatically acquire high-resolution reference data for low-resolution DEM data using the self-similarity of terrain data.

To validate the accuracy of the model, we conducted three sets of experiments on experimental data selected from different terrain types: Inner Mongolian Plateau, Qinling Mountains, Tarim Basin, and North China Plain. The first set of experiments aimed to verify the accuracy of the model presented in this study. The experimental results showed that the model achieved more than 90% SSIM values in all four areas, which demonstrated the high accuracy of the model in reconstruction. The second set of experiments is compared with bicubic interpolation, SRGAN, and SRCNN methods to verify the reconstruction quality of the model proposed in this paper. The comparison results showed that in gentler terrain, SSTrans had the best reconstruction effect but was not outstanding. In more complex terrain, SSTrans shows a significant improvement in reconstruction compared to other methods, with indexes notably higher. The third set of experiments evaluates the terrain attributes (slope and aspect). In areas 1 and 2, SSTrans does not show a significant advantage over SRCNN in the reconstruction of elevation values, but it does demonstrate significant improvement in slope and aspect. In areas 3 and 4, which are two areas with more complex terrain surfaces, the reconstruction of SSTrans is more outstanding. SSTrans, a reference-based image super-resolution (RefSR) method, is able to produce more accurate results when compared to SRGAN and SRCNN, two single-image super-resolution (SISR) methods based on deep learning. This is because it uses reference images obtained through self-similarity to gather more specific data when dealing with complex terrain surfaces.

In future work, we will further attempt to introduce adversarial generative network methods in combination with SSTrans methods to investigate how to further improve the reconstruction accuracy.

**Author Contributions:** Conceptualization, X.Z.; methodology, X.Z. and Z.B.; software, Z.B.; validation, X.Z., Z.B. and Q.Y.; formal analysis, Z.B.; data curation, Z.B.; writing—original draft preparation, Z.B. and X.Z.; writing—review and editing, Z.B. and Q.Y.; visualization, Z.B.; supervision, Q.Y.; project administration, Q.Y. and X.Z.; funding acquisition, Q.Y. and X.Z. All authors have read and agreed to the published version of the manuscript.

**Funding:** The research work described in this paper was supported by the Joint Research Fund in Astronomy (U2031136) under a cooperative agreement between the NSFC and CAS.

**Data Availability Statement:** The data were obtained from https://search.earthdata.nasa.gov/search/ (accessed on 3 January 2023).

**Acknowledgments:** The authors would like to thank the reviewers for their constructive comments and suggestions and Ziyi Chen for her valuable discussions.

**Conflicts of Interest:** The authors declare no conflict of interest.

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
