# Peer review of "Terrain Self-Similarity-Based Transformer for Generating Super Resolution DEMs"

_remotesensing, doi:10.3390/rs15071954_

Round 1

Reviewer 1 Report

I think this work is valuable, but there are some issues that need to be clarified

1、super-resolution (SISR) techniques are used to get high resolution DEM, how about the results of the reconstruction DEMs in the paper

2、DEM is used for terrain analysis. I think this manuscript shoud evaluate the terrain attributes (slope\aspect\gully, etc.)  of the oringinal and reconstruction DEMs

3. fig.5 and fig.6 need scale ruler

Reviewer 2 Report

This study proposed a method called SSTran, which is a transformer-based super-resolution method to generate high-resolution DEM. According to the experiment results, the proposed method performs well, especially in complex and uneven terrain areas, and also can achieve superior results than commonly used SR methods. Actually, the topic of DEM super-resolution is of great importance, however, there are many considerable improvements/clarifications needed, including the novelty of this study, inadequate introduction, and absence of key information in methodology and results. Therefore, I recommend the manuscript can not publish at this stage and should make a careful revisions.

Major comments:

1. In Introduction section, the authors need to define the importance and the novelty of their work more clearly. Specifically, authors need to claim why the super-resolution of DEM is important. I think the related content of the first paragraph is not adequate. And I believe authors need to mention which resolution is needed for those researches and how high of the resolution is appropriate. Moreover, authors need to describe the terrain self-similarity more specifically in the introduction section, since this is perhaps the biggest novelty of this work. For example, what is terrain self-similarity, and why the similarity is good for the super-resolution of DEM?

2. In Methodology section, please provide more information about the caption of Figure 1. What does the DEM-LR mean? As well as the DEM-HR, DEM-Ref. Also the meaning of the up and down arrow. Although the related contents are already provided in line 98 to 114, to improve the readability, these statements still need to be mentioned in the image caption. The same problem also appeared in Figure 2 and 3.

3. Another confusing point is the spatial resolution of the reconstructed DEM after super-resolution. From 3.1 section, this work used the ASTER GDEM as the experiment data, which is a 30 m resolution DEM. So what is the spatial resolution of the SR DEM? I believe this is a piece of significant information that should be provided.

4. In Figure 5, a1 and b1, a2 and b2, a3 and b3, a4 and b4, those sub-figures look so similar that it is hard to tell the difference. I recommend authors can use the hillshade instead of compare the elevation directly. Because hillshade can reflect the terrain relief more clearly than elevation. The problem also occurs in figure 6.

5. There is another issue needs to be taken by the authors. DEM is different from natural images. DEM can provide terrain surface information and usually be used as a fundamental dataset in many topographic-based kinds of research. Therefore, only using accuracy metrics to evaluate the performance of DEM super-resolution is not enough. Authors should compare the original DEM and reconstructed DEM directly in a specific application. For example, authors can compare the difference of the elevation histogram between the original DEM and the SR DEM. Also, the difference of the drainage network extraction results between the original DEM and SR DEM can be compared as well.

Reviewer 3 Report

The authors of the publication addressed the current problem of generating super-resolution DEMs. The topic taken is interesting. The work does not raise any objections in terms of its content. The form of the research and the analyzes carried out are appropriate. The proposed methodology is appropriate. It is worth considering developing and testing the proposed method on other research plots.

Reviewer 4 Report

It is a very interesting investigation in the field of Production of Digital Elevation Models.

The work seems very acceptable to me as they propose and develop it, congratulations.

I only indicate that the conclusions have seemed few to me

Round 2

Reviewer 1 Report

I agree with the authores' responses.

Reviewer 2 Report

I am very appreciate authors effort to address most issues raised from the review comments. However, I still have some concerns about this work.

(1) The author mentioned that the resolution of both original DEM and super-resolution DEM are 30m. That makes me confused. There are many kinds of open-access DEMs with 30m resolution. Such as the SRTM, ALOS AW3D30, NASADEM, etc. Since the 30m resolution DEMs are easy to acquire, what is the significance of this work? I think authors could attempt to design a framework to generate a super-resolution DEM with the resolution higher than 30m. Therefore, I think a more reasonable way to carry out the experiment is to choose a DEM with the resolution higher than 30m as the HR-DEM, and the 30m DEM or 90m DEM as the LR-DEM, and then reconstruct DEM based on your proposed method.

(2) Although the revised manuscript added the comparing analysis about slope and aspect extracted from original DEM and super-resolution DEM. However, I still insist that author need to extract the stream network or other terrain features (such as mountain ridge line, gully boundary, etc) from original DEM and reconstructed DEM, respectively. And compared the extracted results. Because DEM is a fundamental datatset to make terrain analysis. Therefore, comparing the terrain features extracted from both DEMs is very important.

(3) Another minor comment is about the Figure 8. I think the hillshade results are not correct. DEMs need to project from the geographic coordinate system to the projected system first and then the hillshade can be calculated correctly.